# Reduced Plasma Guanylin Levels Following Enterotoxigenic *Escherichia coli*-Induced Diarrhea

**DOI:** 10.3390/microorganisms11081997

**Published:** 2023-08-03

**Authors:** Ingeborg Brønstad, Hilde Løland von Volkmann, Sunniva Todnem Sakkestad, Hans Steinsland, Kurt Hanevik

**Affiliations:** 1National Centre for Ultrasound in Gastroenterology, Haukeland University Hospital, 5021 Bergen, Norway; ingeborg.bronstad@helse-bergen.no (I.B.); hilde.loland.von.volkmann@helse-bergen.no (H.L.v.V.); 2Department of Medicine, Haukeland University Hospital, 5021 Bergen, Norway; 3Department of Clinical Science, University of Bergen, 5021 Bergen, Norway; sunniva.sakkestad@gmail.com; 4National Center for Tropical Infectious Diseases, Department of Medicine, Haukeland University Hospital, 5021 Bergen, Norway; 5Centre for Intervention Science in Maternal and Child Health (CISMAC), Centre of International Health, Department of Global Public Health and Primary Care, University of Bergen, 5020 Bergen, Norway; hans.steinsland@uib.no; 6Department of Biomedicine, University of Bergen, 5020 Bergen, Norway

**Keywords:** ETEC, experimental infection, guanylin, uroguanylin, zinc, heat-stable toxin, diuresis

## Abstract

The intestinal peptide hormones guanylin (GN) and uroguanylin (UGN) interact with the epithelial cell receptor guanylate cyclase C to regulate fluid homeostasis. Some enterotoxigenic *Escherichia coli* (ETEC) produce heat-stable enterotoxin (ST), which induces diarrhea by mimicking GN and UGN. Plasma concentrations of prohormones of GN (proGN) and UGN (proUGN) are reportedly decreased during chronic diarrheal diseases. Here we investigate whether prohormone concentrations also drop during acute diarrhea caused by ST-producing ETEC strains TW10722 and TW11681. Twenty-one volunteers were experimentally infected with ETEC. Blood (*n* = 21) and urine (*n* = 9) specimens were obtained immediately before and 1, 2, 3, and 7 days after ETEC ingestion. Concentrations of proGN and proUGN were measured by ELISA. Urine electrolyte concentrations were measured by photometry and mass spectrometry. Ten volunteers developed diarrhea (D group), and eleven did not (ND group). In the D group, plasma proGN, but not proUGN, concentrations were substantially reduced on days 2 and 3, coinciding with one day after diarrhea onset. No changes were seen in the ND group. ETEC diarrhea also seemed to affect diuresis, the zinc/creatinine ratio, and sodium and chloride secretion levels in urine. ETEC-induced diarrhea causes a reduction in plasma proGN and could potentially be a useful marker for intestinal isotonic fluid loss.

## 1. Introduction

The human intestine has a daily total fluid load of about 9 L, where only approximately 10% derives from ingested food and beverage, and the remaining fluid load derives from secretions of the salivary glands, stomach, pancreas, and biliary systems, as well as the intestine itself [1]. Normally, the intestine has a good capacity for absorbing fluid, with the small intestine absorbing about 6–7 L daily, 1.5–1.9 L are absorbed by the colon, and only about 0.1–0.5 L remain in the stool [2]. Balanced intestinal fluid secretion is necessary for maintaining appropriate gut fluidity and lubrication.

The endogenous peptide hormones guanylin (GN) and uroguanylin (UGN) are effectors involved in regulating fluid and ion homeostasis in the intestine [3,4]. GN (15 amino acids (aa)) and UGN (16 aa), commonly referred to as the guanylins, are secreted into the lumen after C-terminal enzymatic cleavage of their short prohormones, called proGN (94 aa) and proUGN (86 aa), respectively [5,6,7,8,9]. The hormones act through binding to intestinal guanylate cyclase C (GC-C) receptors, which are found within the apical membranes of epithelial cells and on basolateral epithelial surfaces and are expressed from the duodenum to the rectum [10,11]. Activation of GC-C results in increased secretion of Cl^−^ and HCO_3_^−^ and reduced absorption of Na^+^, which results in an increased fluid load into the intestinal lumen. Animal experiments suggest they may also have a natriuretic effect in the kidney [12].

The sequence homology of the active peptides GN and UGN is about 50%, and both peptides have two disulfide bonds between cysteine residues, which are essential for their activities [5,7,13]. However, the structural differences make their activity pH-dependent, where UGN is more potent in an acidic environment, while GN is more potent at a higher pH [14,15]. This is also reflected in the primary sites of production of the two hormones, where UGN is mostly expressed in the duodenum and jejunum and GN in the distal small intestine and colorectum [16].

Excessive secretion and/or impaired fluid and electrolyte absorption across the intestinal epithelium may result in diarrhea. Enterotoxigenic *Escherichia coli* (ETEC) may cause diarrhea by producing heat-stable enterotoxin (ST) and/or heat-labile toxin (LT). ETEC is an important cause of diarrheal disease in young children living in low- and middle-income countries and travelers to these countries [17,18]. Interestingly, ST produced by ETEC induces diarrhea by binding to the GC-C receptor with high affinity [19], as it is structurally similar to and can mimic the action of GN and UGN in the intestine. ST consists of 19 aa, is about 44% and 63% homologous to GN and UGN, respectively, and folds by forming three disulfide bonds [5,6,7,20].

The proGN and proUGN prohormones are released into the bloodstream and are detectable in plasma [21], but their systemic endocrine function is not well known [22]. Previous studies have shown that plasma proGN and proUGN concentrations tend to be reduced during chronic diarrheal diseases, such as Crohn’s disease and Familial *GUCY2C* Diarrhea Syndrome (FGDS) loss [23,24]. This suggests that plasma proGN and proUGN could be useful markers reflecting gut GN and UGN concentrations and thereby intestinal fluid balance. In addition, studies based on HPLC methods and cGMP bioassays have shown that bioactive UGN is present in urine [4,7,25], while GN is not [25].

This study aimed to assess whether the levels of guanylin prohormones are reduced during acute diarrhea caused by ST-producing ETEC strains, similarly to what has been seen in chronic diarrhea patients. To do this, we monitored the change in plasma proGN and proUGN concentrations in volunteers who were experimentally infected with ETEC strains that produce ST but not LT. We also evaluated diuresis, urine electrolyte concentrations, and the secretion of guanylins in urine in a subgroup.

## 2. Materials and Methods

### 2.1. Experimental Infection Study

The present study is based on data and specimens from two studies where 30 healthy volunteers were experimentally infected with ETEC, which produces the human variant of ST (STh) but not LT [26,27,28]. The present study is based on data and specimens from all 9 volunteers who were infected with strain TW11681 and 12 who were infected with strain TW10722. The overall objective of the experimental infections was to develop a challenge model to test ST-based toxoid vaccine candidates. Strain TW11681 did not consistently produce diarrhea, but this was achieved with strain TW10722. The 21 volunteers were students aged between 19 and 29, 19 (90%) of whom were female. The enrollment process, inclusion and exclusion criteria, and medical screening tests have been described in detail earlier [26,27]. In brief, these studies took place in October/November 2016, September 2017, and March 2018 at the Infectious Diseases Ward at the Division for Infectious Diseases at Haukeland University Hospital (HUS), Bergen, Norway. The volunteers signed an informed consent document and were enrolled after they had been through medical screening tests and questionaries to make sure they understood the rationale and the requirements of the study. The experimental infection was performed by oral ingestion of doses between 1 × 10^6^ and 1 × 10^10^ colony-forming units after an overnight fast.

### 2.2. Specimen Collection

Blood specimens and overnight urine were collected in the morning of the screening day (ds), which was 20 to 48 days before ETEC ingestion, immediately before ingesting the ETEC dose (day 0), and at days 1, 2, 3, and 7 after ETEC ingestion.

The blood specimens were drawn into pre-cooled EDTA tubes by venipuncture and kept on ice for a maximum of 30 min before centrifugation at 1800× *g* for 10 min at 4 °C. The resulting plasma was aliquoted and stored at −80 °C until analysis. All blood specimens were obtained after at least 7 h of fasting, as none of the volunteers became ill enough to need oral or intravenous fluids during the night.

Urine sample collection was not part of the original sampling protocol and was introduced only after volunteers tolerated added sampling on top of the quite extensive sampling regimen, and ethics approval was obtained for this expansion. Therefore, we could collect urine from only 9 volunteers who were infected with TW10722, and for practical reasons, we collected overnight urine specimens each morning. Timepoints for last urination before sleep and first time after sleep, as well as the total urine volume between last urination before sleep and first time after sleep, were recorded to determine overnight diuresis, defined as mL urine/hour. The urine specimens were kept in a refrigerator until centrifugation at 1000× *g* for 10 min at 4 °C, followed by aliquoting the supernatant and storage at −80 °C until analysis. Analyte urine secretion was calculated by multiplying analyte concentration, as determined as described below, with urine volume, divided by hours from the last urination before sleep to the first urination after.

### 2.3. Clinical Evaluation

The volunteers were divided into a diarrhea (D) group and a non-diarrhea (ND) group based on the development of diarrhea, which was defined as the passing of 1 loose/liquid stool of ≥300 g, or ≥2 loose/liquid stools totaling ≥200 g during any 48 h period within 120 h after ingestion of the ETEC dose [26,27]. The diarrhea episode was graded as mild if the volunteer produced 1–3 loose stools totaling 200–400 g per 24 h, moderate with 4–5 loose stools or a total weight of 401–800 g per 24 h, and severe with ≥6 loose stools or a total weight of >800 g per 24 h [26,27].

### 2.4. Biochemical Analyses

ProGN and proUGN concentrations in plasma and urine were quantified by using the Human Proguanylin ELISA and the Prouroguanylin Human ELISA kits (Cat. Nos. RD191046100R and RD191069200R, respectively; BioVendor a.s., Brno, Czech Republic). The ELISAs were performed according to the manufacturer’s instructions. The ELISA plate optical readings were done by using a SPECTRAmax microplate reader (Molecular Devices, Sunnyvale, CA, USA), and concentrations were calculated by using SoftMaxPro Software version 7.1 (Molecular Devices, Sunnyvale, CA, USA) using a curve fit based on its 4-parameter logistic nonlinear regression model.

Urine chloride, sodium, potassium, and protein levels were measured using the Cobas^®^ c 702 module (Roche, Basel, Switzerland); urine creatinine was measured using a AU680 Clinical Chemistry Analyzer (Beckman Coulter, Brea, CA, USA); and urine zinc was measured using inductively coupled plasma mass spectrometry (ICP-MS) on an ELAN DRC-e (PerkinElmer Inc., Waltham, MA, USA), all performed by the Laboratory Clinic at HUS.

### 2.5. Statistical Analyses

We tested for differences in proGN, proUGN, overnight diuresis, and electrolytes between the D and ND groups for each given sampling day by using Mann–Whitney U tests. The Friedman test, followed by Dunn–Bonferroni post hoc tests for multiple comparisons, was used to identify significant changes in plasma and urine concentrations between before dose ingestion (day 0) and the days after (days 1, 2, 3, and 7), while Wilcoxon paired tests were used to test the changes in concentrations between day 0 and the days of diarrhea onset and the day after the development of diarrhea. Spearman’s rank-order tests were used for correlation analyses. We considered *p* values of <0.05 to be statistically significant.

### 2.6. Ethical Statement

All volunteers gave informed written consent to participate in this study, which was approved by the Regional Committee for Medical and Health Research Ethics, Health Region West (Project ID: 2014-826) on 13th June 2014 and registered at ClinicalTrials.gov (Project ID: NCT02870751).

## 3. Results

### 3.1. Clinical Evaluations

Two (22%) of the nine volunteers ingesting the TW11681 strain and eight (67%) of the twelve volunteers ingesting the TW10722 strain developed diarrhea (Table 1).

### 3.2. Plasma proGN and proUGN

The change in plasma proGN and proUGN concentrations over time for all volunteers is shown in Figure 1, and median values (25th, 75th percentiles) are given in Appendix A.

D group volunteers had a significant decrease in median plasma proGN concentration following dose ingestion, from 7.20 ng/mL on day 0 to 5.71 ng/mL on day 2 (*p* = 0.008) and to 5.64 ng/mL on day 3 (*p* = 0.012). No such changes in plasma proGN concentrations were seen in the ND group. In a day-by-day comparison between the D and ND groups, the median proGN concentrations only differed significantly on day 3 (5.64 and 6.49 ng/mL, respectively; *p* = 0.005) (Figure 1a,b; Appendix A).

Neither the D nor ND group volunteers had clear changes in median proUGN concentration between day 0 and any of the follow-up days. The median proUGN concentrations did not seem to differ substantially between the D and ND groups for any of the study days (Figure 1c,d).

To further examine the association between diarrhea and the concentration of plasma guanylins, we compared the median proGN and proUGN concentrations on day 0 with those at diarrhea onset and the day after diarrhea onset (Figure 1a,c). There were no significant differences in the proGN or proUGN concentrations between day 0 and the day of diarrhea onset. However, proGN concentration decreased significantly from a median of 7.20 ng/mL on day 0 to 5.35 ng/mL on the day after diarrhea onset (*p* = 0.002) (Figure 2a). Correspondingly, we observed no significant changes in the proUGN concentration between day 0 and the day of diarrhea onset or the day after (Figure 2b). The observed decrease in proGN concentrations at group level led us to examine whether this decrease correlated with diarrhea severity or diarrheic stool output, as measured in maximum stool weight during any 24 h period of the diarrheal episode (Table 1), and the severity scores (mild, moderate, and severe) as described in clinical evaluation. However, we did not find this to reach significance (Appendix A).

### 3.3. Correlation between Plasma proGN and proUGN Concentrations

Interestingly, there were significant correlations between plasma proGN and plasma proUGN concentrations on day 0 (r = 0.665, *p* = 0.001), day 1 (r = 0.636, *p* = 0.002), and day 7 (r = 0.458, *p* = 0.037), but only weak correlations on days 2 (r = 0.301, *p* = 0.185) and 3 (r = 0.300, *p* = 0.187). When performing these analyses separately for volunteers in the D and ND groups, we found that the weak correlation on days 2 and 3 was an effect of reduced concentrations of plasma proGN among the D group volunteers on these days (Appendix A).

### 3.4. Overnight Diuresis and Urine Analyte Concentrations

For the 9 volunteers who were infected with TW10722, we estimated diuresis, urine protein/creatinine and zinc/creatine ratios, and urine sodium, potassium, and chloride concentrations (Figure 3). The median overnight diuresis was significantly decreased in the D group at day 1 (26.8 mL/h, *p* = 0.014), day 2 (28.6 mL/h, *p* = 0.008), and day 7 (27.4 mL/h, *p* = 0.042) compared to day 0 (52.5 mL/h). Furthermore, diuresis was significantly lower in the D group (28.6 mL/h) compared to the ND group (58.8 mL/h) (*p* = 0.048) at day 2. There were no clear correlations between diuresis and plasma proGN or proUGN concentrations.

Urine protein/creatinine ratios and urine chloride concentrations for all examined specimens were within the normal reference range given by the Laboratory Clinic at HUS. The urine zinc/creatine ratios decreased in both groups on day 3, reaching significance in the D group with 0.32 µmol/mol compared to 0.46 µmol/mol on day 0 (*p* = 0.014). Urine chloride secretion was significantly lower (*p* = 0.024) on day 1 in the D group (70 mmol/L) compared to the ND group (167 mmol/L). Urine sodium secretion was significantly higher on day 1 with 8.24 mmol/h and on day 2 with 4.88 mmol/h for the ND group, compared to the D group with 2.44 mmol/h on day 1 (*p* = 0.024) and 1.96 mmol/h on day 2 (*p* = 0.024) (Figure 3).

### 3.5. Urine proGN and proUGN Secretion Rates

Since concentrations of urine proGN and proUGN will vary depending on the hydration status of the volunteer, for these analyses we compare urine secretion rates rather than concentrations by adjusting for diuresis. After converting the proGN and proUGN urine concentrations to secretion rates by adjusting for overnight diuresis, we found that the rate of proGN and proUGN secretion into urine varied greatly between volunteers, ranging from 40.2 to 719 ng/h for proGN and 1.9 to 379 ng/h for proUGN, as well as between different days for each volunteer (Figure 4). We found no clear indication that the experimental infection itself, the diarrhea, or the severity of the diarrhea affected the secretion rate. There were significant correlations between proGN and proUGN secretion rates in urine on day 1 (r = 0.745, *p* = 0.027) and day 2 (r = 0.733, *p* = 0.031), but not on other days, and there were no clear correlations between plasma and urine proguanylins, except for between plasma proUGN concentrations and urine proUGN secretion rates on day 7 (r = 0.717, *p* = 0.037).

## 4. Discussion

In this study, our aim was to gain more insight into the dynamics of GN and UGN prohormone regulation during acute diarrheal disease caused by experimental infection with ST-producing ETEC. Examining fasting plasma concentrations during the course of the infection, we found a drop in proGN levels, which appeared to be associated with having experienced diarrhea, while the plasma proUGN concentrations seemed unaffected.

It is so far presumed that both GN and UGN have paracrine functions in the intestine as ligands for the GC-C receptor [8]. UGN might also be involved in the regulation of satiety and body weight [29,30]. Previous studies on patients with chronic diarrheal diseases such as Crohn’s disease and FGDS have shown that changes in intestinal hydration levels are reflected in changes in plasma proGN concentrations [23,24]. The results from the present study indicate that this holds true for acute diarrhea caused by ST-producing ETEC. By using ETEC strains that only produce ST but not LT, we ensured that diarrhea was mainly caused by exposure to ST. However, it remains unclear whether it is the direct involvement of ST or diarrhea in general that triggers the changes in plasma proGN concentrations. So far, we can only speculate about the underlying mechanisms for these changes, but in patients with inflammatory bowel disease, which often causes chronic diarrhea, downregulation of the genes encoding proGN, proUGN, and GC-C has been reported [31,32].

Plasma proguanylins have also been reported to decrease after a meal, where plasma proUGN dropped about 30 min after a food intake [24,33], and plasma proGN seemed to drop later, with a peak bottom after 2 h [24]. This delay probably reflects the different locations where GN and UGN primarily act. UGN is mostly expressed and is more active in the upper intestinal tract, where the pH is lower, while GN is more expressed and is more active in the distal small intestine and colon, where the pH is higher [14]. Interestingly, in the present study, plasma proGN and proUGN concentrations correlated well with each other before the volunteers developed diarrhea, but not after the proGN concentrations dropped in the volunteers who developed diarrhea. This indicates a real differential response between these two prohormones, with ETEC-induced diarrhea mostly affecting fluid homeostasis in the distal small intestine and possibly the colon [34,35].

A non-significant rise in plasma proGN was seen on day 1 in several volunteers from both groups. This could potentially be explained by ETEC-produced ST being taken up into the circulation and cross-reacting with ELISA kit antibodies. However, due to the non-invasive nature of ETEC and the difference in aa sequence, we believe this is unlikely.

Recent data suggest that neuropod cells in the intestinal mucosa, with high numbers of GC-C receptors, could specifically mediate the reduction in visceral pain by oral GC-C agonists seen in constipation syndromes [36]. However, there were no signs of any analgesic effect in volunteers exhibiting diarrhea. Abdominal cramps and discomfort were well correlated with the development of diarrheal disease as well as ETEC proliferation, as previously published [28,37]. It is likely that the mucosal inflammation, elicited by a non-invasive pathogen such as ETEC and leading to visceral pain perception, overrides any potential analgesic effect of ST being a GC-C agonist.

As could be expected, the overnight diuresis decreased when the volunteers developed diarrhea. The gold standard for determining hydration status is the direct measurement of plasma osmolality, but this is not a reliable method to evaluate isotonic dehydration, which often occurs during episodes of acute secretory diarrhea [38]. Our results indicate that plasma proGN could be a useful marker to assess gut-related fluid loss, and further studies to assess its suitability are warranted.

It is well known that zinc deficiency leads to increased susceptibility to enteric infections [39] and that zinc supplementation can reduce the severity of such infections, for example, by increasing intestinal concentrations of protective metallothionein [40]. The zinc-bound metallothionines may, for example, help detoxify ST through reduction of the disulfide bonds [41], and the increased secretion of oxidized free zinc in the urine of the D group volunteers indicates that this type of protective mechanism has been active in these volunteers. The finding that all volunteers, including the ND group volunteers, had reduced zinc/creatine ratios on day 3 indicates that the ETEC infections contributed to depleting zinc stores in all these volunteers, irrespective of diarrhea.

ETEC diarrhea did not seem to have any large effect on the concentrations of sodium and chloride in urine, although their concentrations in the D group were comparatively lower than in the ND group both on days 1 and 2. These results are puzzling, but it is possible that the D group may have lost excess salts mainly via the intestine, while the ND group lost them through urine.

### Limitations

In this study, the volunteers developed relatively mild diarrheal disease and were in generally good health even when reaching the study’s definition of severe diarrhea. This is reflected in the significant but relatively minor changes in plasma proGN and the relatively small changes in urine electrolytes.

We collected urine samples from only nine volunteers in this study, so the power to do statistical analysis on the urine data is limited. We also collected the urine only overnight, and we did not register or compensate our measurements for fluid intake. Employing 24 h diuresis and fluid intake records would have given us a better basis for assessing changes in diuresis and fluid balance.

The ELISA kits used to detect proGN and proUGN were coated with polyclonal anti-human proguanulin/prouroguanylin antibodies, and the standard for quantification was recombinant proguanylin/prouroguanylin (11.5 and 10.7 kDa proteins, respectively). Since the mature hormones are cleaved from and, therefore, share a part of the amino acid sequences with their respective prohormones, we cannot exclude the possibility that part of the measured signal may be due to mature guanylin peptides in plasma.

To our knowledge, proGN has not been detected in urine before [25], and the ELISA kits used to detect proGN and proUGN have not been properly validated for use on urine specimens. Recognizing that ELISA kit antibodies could be mainly targeting epitopes on the proguanylins, which are unlikely to reach the urine, the actual urine proGN and proUGN secretion rates estimated in this study are uncertain.

## 5. Conclusions

We found that fasting plasma proGN concentrations dropped after experiencing acute diarrhea caused by ST-producing ETEC. Plasma proGN could be a useful marker of isotonic intestinal fluid loss, although further investigations are needed to evaluate whether it is specific only to diarrhea caused by infection with ST-producing ETEC.

## Figures and Tables

**Figure 1 microorganisms-11-01997-f001:**
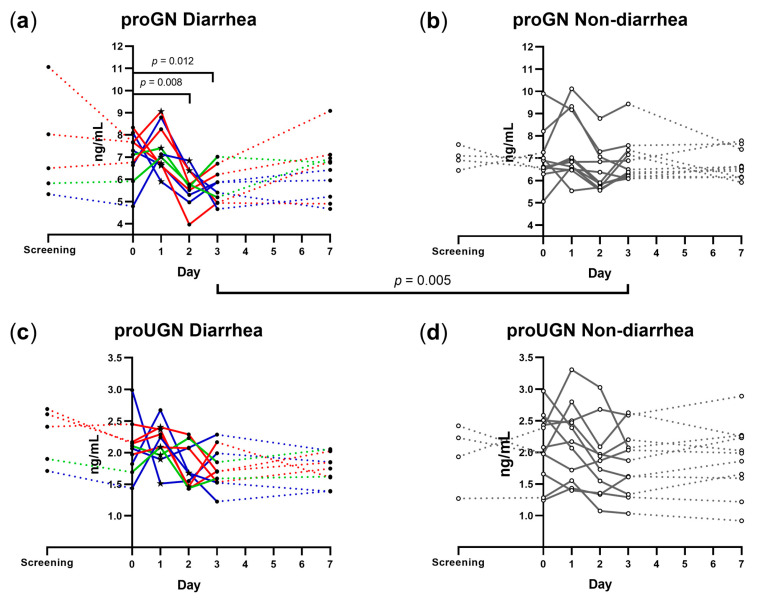
Plasma proguanylin (proGN) (**a**,**b**) and prouroguanylin (proUGN) (**c**,**d**) concentrations in experimentally infected volunteers who did (**a**,**c**) and did not (**b**,**d**) develop diarrhea, measured at screening day, immediately before ETEC ingestion (day 0), and days 1, 2, 3, and 7 days after ETEC ingestion. A star (★) indicates the onset of diarrhea. Red lines: severe diarrhea; green lines: moderate diarrhea; blue lines: mild diarrhea; grey lines and hollow circles: no diarrhea. Dotted line is used when samples are not taken on consecutive days. Open circles: non-diarrhea group; solid circles: diarrhea group A horizontal bar displaying the *p* value is used to indicate significant differences in median concentrations between days or groups.

**Figure 2 microorganisms-11-01997-f002:**
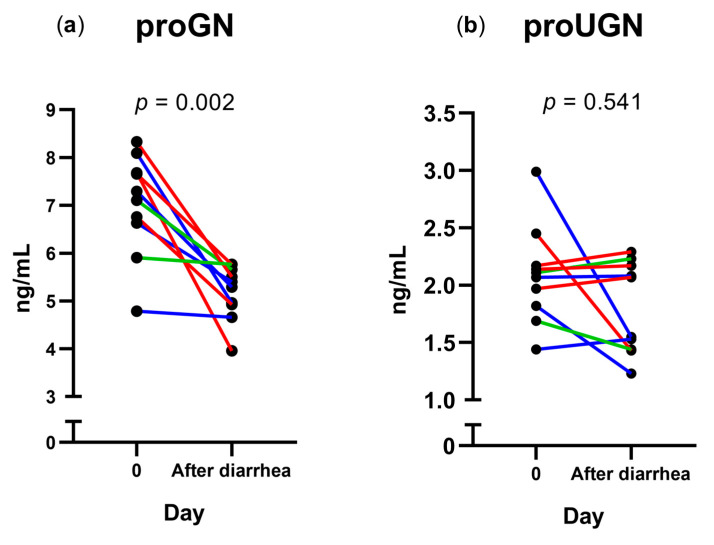
Changes in plasma (**a**) proguanylin (proGN) and (**b**) prouroguanylin (proUGN) concentrations between immediately before ETEC ingestion and the day after diarrhea onset among 10 volunteers who developed diarrhea. Red lines: severe diarrhea; green lines: moderate diarrhea; blue lines: mild diarrhea. Solid circles: diarrhea group.

**Figure 3 microorganisms-11-01997-f003:**
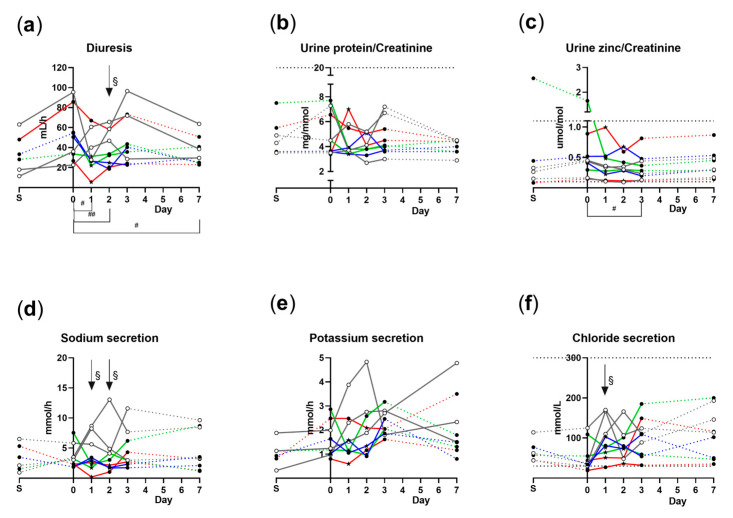
Urine analyses from nine healthy volunteers at a screening day, immediately before ingestion of ETEC (day 0), and days after ETEC ingestion (days 1, 2, 3, and 7). (**a**) Diuresis; (**b**) urine protein/creatinine ratio; (**c**) urine zinc/creatinine ratio; (**d**) sodium secretion; (**e**) potassium secretion; (**f**) urine chloride secretion. A star (★) indicates the onset of diarrhea. Open circles: non-diarrhea group; solid circles: diarrhea group. Red lines: severe diarrhea; green lines: moderate diarrhea; blue lines: mild diarrhea; grey lines: no diarrhea. Dotted line is used when samples are not taken on consecutive days. Dashed horizontal lines in (**b**,**c**,**f**) indicate the upper reference cutoff for normal values. #: *p* < 0.05 between days for the diarrhea group. ##: *p* < 0.01 between days for the diarrhea group. §: *p* < 0.05 between diarrhea and non-diarrhea groups.

**Figure 4 microorganisms-11-01997-f004:**
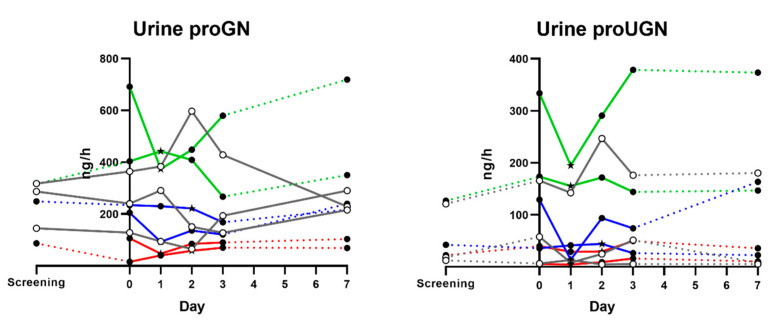
Rate of proGN and proUGN secretion in urine samples from nine volunteers. Values are based on concentrations as measured by ELISA immunoreactivity in urine, adjusted for diuresis. A star (★) indicates the onset of diarrhea. Open circles: non-diarrhea group; solid circles: diarrhea group. Red lines: severe diarrhea; green lines: moderate diarrhea; blue lines: mild diarrhea; grey lines with open symbols: no diarrhea. Dotted line is used when samples are not taken on consecutive days.

**Table 1 microorganisms-11-01997-t001:** Clinical outcomes after ETEC ingestion.

	No. of Volunteers (Females)	Mean Age, (SD) (Years)	Strain Used, No. of Volunteers		Mean Diarrheal Stool Output/24 h (Grams)	Mean Time Until Diarrhea (h)
			TW 11681	TW 10722		
Non-diarrhea (ND)	11 (10)	24.5 (3.2)	7	4	Not applicable	Not applicable
Diarrhea (D) (all)	10 (9)	23.5 (1.4)	2	8	472	33
Mild	4 (4)	24.2 (2.1)	2	2	375	42
Moderate	2 (1)	23.5 (0.0)	0	2	465	25
Severe	4 (4)	22.8 (0.5)	0	4	572	28

## Data Availability

Most of the data acquired in this study are presented in the manuscript and Appendix A. Raw data supporting the findings of this study are available from the corresponding author upon reasonable request.

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
