# Peer review of "Reduced Plasma Guanylin Levels Following Enterotoxigenic Escherichia coli-Induced Diarrhea"

_microorganisms, 2023, doi:10.3390/microorganisms11081997_

Round 1

Reviewer 1 Report

This study by Brønstad et al. titled “Reduced plasma guanylin levels following enterotoxigenic Escherichia coli-induced diarrhea” reports that circulating prohormone guanylin titers decrease in volunteers in which ETEC infection develops into secretory diarrheal disease, as compared to non-secretory diarrheal disease. These data are important because they suggest that ETEC infection can causes changes in circulating hormones, which expands the reach of this enteric pathogen beyond the mucosae to systemic functions. Therefore, these data provide clues as to how ETEC-mediated secretory diarrhea may affect other aspects of ETEC morbidity including growth stunting and impaired cognitive development. The authors also demonstrate that diarrheal biomarkers may be able to be identified in non-stool samples (e.g., urine), where urine of volunteers that developed secretory diarrhea had lower chloride that the urine of volunteers that did not develop secretory diarrhea. These data are also important since overall fluid homeostasis and hydration/dehydration status must consider total fluid loss including those diverted from diuresis towards secretory diarrhea. These findings will be important to a broad audience but will be especially important to those interested in understanding how enteric pathogens alter human physiology that leads to disease.

 Major concerns: 

The authors do not discuss how the prohormones are differentiated from the mature hormones in the ELISA.  Therefore, it is uncertain if the levels of “prohormones” is actually “prohormones + mature hormones”. Please clarify.

 In Figure 1, it is difficult to see individual the trends in proGN following infection since the data and trendlines overlap. However, it appears as though there are increases in plasma proGN titers in at least 6 out of the 10 volunteers from day 0 to day 1. Is this statistically significant?  Does this mean that circulating proGN titers increase before they decrease?  Does this suggest that they are increased to block diuresis in the kidney/bladder since? Also, would that mean that ETEC infection increases proGN levels, but ETEC pathogenesis (e.g., diarrhea) causes decreased serum proGN? Please comment.

 Supplemental figure 2.

What happens if the plasma data is broken down and compared based on severity of diarrhea? Also, if urine proGN and proUGN were binned into high-, medium-, and low-level outputs, would there be a difference in proGN and proUGN secretions in volunteers that developed mild, moderate, and severe diarrhea?

 Could it be that ST produced by ETEC makes its way into the sera and cross reacts with antibodies in the proGN ELISA?  Can this be ruled out?

 The authors should incorporate some discussion to recent studies that demonstrate that GC-C may be a sensor of visceral pain.  Barton et al. 2023, JCI, PMID 36548082

 Minors:

Lines 2, 45, 73, 172, 185, 218, 243 spelling, guanyline to guanylin.

Line 24, identify ETEC strain.

Line10, GC-C is also present on basolateral epithelial surfaces. Albano et al. 2001, BBRC, 284, 3, 331-4

Line 198. Table 1 indicates that 12 volunteers were infected with TW10722, 8 developing diarrhea and 4 not developing diarrhea. How was the number 9 arrived at in line 198? What do the open and closed symbols mean in Supplemental Figure 1?

 Line 199, zink to zinc and in supplementary

Line 268, not through zinc oxidation, but through reduction of the ST disulfide bonds.

 Author Response

Major concerns: 

  1. The authors do not discuss how the prohormones are differentiated from the mature hormones in the ELISA.  Therefore, it is uncertain if the levels of “prohormones” is actually “prohormones + mature hormones”. Please clarify.

Answer: We thank the reviewer for an important consideration. We used the ELISA kits Human Proguanylin and Human Prourogunylin provided from BioVendor a.s., Brno, Czech Republic. According to the manufacturer, the ELISA plates are coated with polyclonal anti-human proguanulin/prouroguanylin antibodies, and the standard for quantification are recombinant proguanylin/prouroguanylin (11.5 and 10.7 kDa proteins). Since the mature hormones are cleaved from, and therefore share a part of the amino acid sequences with their respective prohormones, we agree that it is possible that the polyclonal antibodies will bind to the short mature hormone peptides. In contrast to more consistent values in plasma, there were large individual variation between and within days of the prohormones measured in urine, which may indicate that the ELISA kits did react to cleaved prohormones.  We have included some sentences about this as a limitation of the study in the discussion.

  1. In Figure 1, it is difficult to see individual the trends in proGN following infection since the data and trendlines overlap. However, it appears as though there are increases in plasma proGN titers in at least 6 out of the 10 volunteers from day 0 to day 1. Is this statistically significant?  Does this mean that circulating proGN titers increase before they decrease?  Does this suggest that they are increased to block diuresis in the kidney/bladder since? Also, would that mean that ETEC infection increases proGN levels, but ETEC pathogenesis (e.g., diarrhea) causes decreased serum proGN? Please comment.

Answer: We agree that there is an interesting trend in the data. It is correct that 6 out of the volunteers that developed diarrhea increased the plasma proGN titers from day 0 to day 1, however, this was not statistically significant (p = 0.5, paired test). Interestingly, the same trend was found for 7 out of 11 volunteers that did not develop diarrhea. It could suggest that ETEC ingestion may increase plasma proGN before a downregulation by the diarrhea.  However, we did not want to speculate too much into this finding given the low number of volunteers. The possibility that this may be caused by uptake of ST into the blood stream and cross reactivity of the ELISA kit with ST cannot be 100% excluded, but is not likely, se response in point 4. We have added this consideration in the discussion.

  1. Supplemental figure 2. What happens if the plasma data is broken down and compared based on severity of diarrhea? Also, if urine proGN and proUGN were binned into high-, medium-, and low-level outputs, would there be a difference in proGN and proUGN secretions in volunteers that developed mild, moderate, and severe diarrhea?

Answer: We found no correlation between severity of diarrhea output and changes in proGN and proUGN. Because of low number (n = 2 for moderate diarrhea) we did not test for statistical differences in the three different severity groups separately. Similarly, for proGN and proUGN in urine, we deemed the number of volunteers to be too small for statistical analysis. Shown in supplementary figure 2, we could not see any trend in proGN and proUGN levels in urine based on severity of diarrhea, where the 2 volunteers with moderate diarrhea had the highest levels and the volunteers with no diarrhea had values similar with mild and severe diarrhea.

To help visualize this, we have included a new figure in the supplementary which shows the change in proGN and proUGN versus total stool output on the day of diarrhea for volunteers in the diarrhea group. For comparison also the comparable total stool output and delta proGN and proUGN values at day 2 for the group not developing diarrhea are included. Although one may see a trend towards lower delta-proGN values with increasing stool output, the data are not conclusive. This is likely due to variablility in colonic fluid reabsorbtion, fluid intake, time from diarrhea until blood sampling, relatively mild diarrhea in the volunteers, and too few volunteers with to even out these factors.

  1. Could it be that ST produced by ETEC makes its way into the sera and cross reacts with antibodies in the proGN ELISA?  Can this be ruled out?

Answer: This is an interesting consideration. We cannot really exclude that the small ST peptide possibly could cross an stressed epithelial border.  However, the homology of ST and mature GN amino acid sequence is about 43 %, and ST and UGN share about 63% homology, whereas the homology between GN and UGN mature hormones are 50%. According to the manufacturer of the proGN ELISA kit, there is no cross-reactivity against human proUGN and UGN Similarly, there is no detectable cross-reactivity for proUGN ELISA against human proGN. Moreover, the proGN ELISA kit has been tested for cross-reactivity by the manufacturer in serum from different mammalian species, including mouse and rat that share more than 93% homology of the mature sequence, resulting in no cross-reactivity. We think therefore that cross reactivity between ST and proGN ELISA is unlikely. We have added a paragraph about it in the discussion.

  1. The authors should incorporate some discussion to recent studies that demonstrate that GC-C may be a sensor of visceral pain.  Barton et al. 2023, JCI, PMID 36548082

Answer: We agree that this could be mentioned in the discussion. Recent data to suggest that neuropod cells in the intestinal mucosa, with high numbers of GC-C receptors, could specifically mediate the reduction in visceral pain by oral GC-C agonists seen in constipation syndromes. However, there were no signs of any analgesic effect in volunteers exhibiting diarrhea. Abdominal cramps and discomfort was well correlated with development of diarrheal disease, as well as ETEC proliferation, as previously published. It is likely that the mucosal inflammation, elicited by a non-invasive pathogen like ETEC, and leading to visceral pain perception, overrides any such effect of the heat-stable toxin being a GC-C agonist. 

Minors:

Lines 2, 45, 73, 172, 185, 218, 243 spelling, guanyline to guanylin.

Answer: We thank the reviewer for useful corrections of spelling. We have now corrected this.

Line 24, identify ETEC strain.

Answer: ETEC strain identifiers have now been given

Line10, GC-C is also present on basolateral epithelial surfaces. Albano et al. 2001, BBRC, 284, 3, 331-4

Answer: We thank the reviewer for this information and have now added this to the text and the reference.

Line 198. Table 1 indicates that 12 volunteers were infected with TW10722, 8 developing diarrhea and 4 not developing diarrhea. How was the number 9 arrived at in line 198? What do the open and closed symbols mean in Supplemental Figure 1?

Answer: We only had urine samples from 9 volunteers, and they were all infected with the TW10722 strain. We have now added the explanation for this under the section of Specimen collection in the Material and methods chapter.

Open symbols mean no diarrhea and closed symbols mean diarrhea. This has now been specified in the figure text.

Line 199, zink to zinc and in supplementary

Answer: Thank you very much, spelling has now been corrected.

Line 268, not through zinc oxidation, but through reduction of the ST disulfide bonds.

Thank you very much. This has now been corrected.

Reviewer 2 Report

I have gone through the Manuscript ID: microorganisms-2496545 entitled: Reduced plasma guanyline levels following enterotoxigenic Escherichia coli – induced diarrhea. The authors investigated if the levels of prohormones proGN and proUGN drop during acute diarrhea induced by E. coli capable of producing heat-stable enterotoxin. Even though research is interesting and novel, some limitations in the study are present, especially when it comes to number of volunteers tested, as authors stated as well. Nevertheless, there are some technical details that should be addressed. I recommend this manuscript for publication after minor revision. My comments are listed below.

 Introduction: Even though the introduction part is well written, I would suggest explaining in more details the connection between proGN and proUGN with E. coli – induced diarrhea.

 Materials and methods: In section 2.1. Experimental infection study (lines 76-80), can you explain why different strains of ST producing E. coli were used to infect volunteers? Why were 12 of them infected with TW10722 strain and 9 of them with TW11681?

In the section 2.2. Specimen collection (lines 99-100), please explain why was overnight urine samples collected only from 9 volunteers infected with TW10722 strain?

 Results: It would be good to put supplementary Figure S1 and Figure S2 in the main text. Important information is presented in these Figures, and at this point only two Figures are presented in the main text.

 Overall, English Language is at moderate level. I would recommend only minor changes, or for native speaker to proofread it. 

Author Response

  1. Introduction: Even though the introduction part is well written, I would suggest explaining in more details the connection between proGN and proUGN with E. coli– induced diarrhea.

Answer: We agree. We have added some more background about the connection between proGN & proUGN with E-coli induced diarrhea, and the similarity between ST and guanylins in the introduction.

  1. Materials and methods: In section 2.1. Experimental infection study (lines 76-80), can you explain why different strains of ST producing E. coliwere used to infect volunteers? Why were 12 of them infected with TW10722 strain and 9 of them with TW11681?

Answer: The two strains were chosen due to the main purpose of the experimental infections being to develop a human challenge mode for testing a toxoid based ETEC vaccine candidate. Both strains were producing heat-stable toxin only, not heat-labile toxin. This is a strength of the study as there should be no heat-labile toxin potentially causing additional fluid loss. Strain TW11681 did not consistently produce diarrhea in the nine volunteers where it was tested, so strain was changed to TW10722 which was found to elicit diarrhea more consistently.   Volunteer numbers are too small for comparison of guanylin levels between strains. In the present manuscript we anyway regard the potential difference between strains to be negligible, as we anticipate the most important outcome is intestinal fluid loss due to diarrhea, and expect there to be a good correlation between ST production and diarrhea. We have added some more explanation about this in the Methods section

  1. In the section 2.2. Specimen collection (lines 99-100), please explain why was overnight urine samples collected only from 9 volunteers infected with TW10722 strain?

Answer: The volunteers were subject to a quite extensive sampling regimen, and we were gradually introducing more sampling dependent on our experience with volunteers tolerance for this. Urine sampling was not introduced before nearing the end of the study, when we saw that there would be room for even more sampling to be performed, and the opportunity for assessing fluid balance parameters could be utilized. When expansion of the sampling regimen was approved by the ethics committee, only 9 remaining volunteers could be included. As we state in the manuscript, we regard the urine examinations to be exploratory.

  1. Results: It would be good to put supplementary Figure S1 and Figure S2 in the main text. Important information is presented in these Figures, and at this point only two Figures are presented in the main text.

Answer: We chose to put these figures into supplementary because of the limitation of power for statistical analysis. However, we agree that this is important exploratory data, and we have now included these figures as part of the main text in the revised manuscript.

  1. Comments on the Quality of English Language

Overall, English Language is at moderate level. I would recommend only minor changes, or for native speaker to proofread it. 

Answer: We have had a native English speaker to proofread the manuscript, and made some minor changes in the revised manuscript, including also corrections suggested by reviewer 1.